# A Mission to Mars: Prediction of GCR Doses and Comparison with Astronaut Dose Limits

**DOI:** 10.3390/ijms24032328

**Published:** 2023-01-24

**Authors:** Ricardo L. Ramos, Mario P. Carante, Alfredo Ferrari, Paola Sala, Valerio Vercesi, Francesca Ballarini

**Affiliations:** 1INFN, Sezione di Pavia, Via Bassi 6, 27100 Pavia, Italy; 2Physics Department, University of Pavia, Via Bassi 6, 27100 Pavia, Italy; 3Institute for Astroparticle Physics, Karlsruhe Institute of Technology, 76021 Karlsruhe, Germany; 4INFN, Sezione di Milano, Via Celoria 16, 20133 Milano, Italy

**Keywords:** space exploration, cosmic rays, astronaut doses, chromosome aberrations, peripheral blood lymphocytes, biomarkers, cell death, relative biological effectiveness, Monte Carlo, biophysical modelling

## Abstract

Long-term human space missions such as a future journey to Mars could be characterized by several hazards, among which radiation is one the highest-priority problems for astronaut health. In this work, exploiting a pre-existing interface between the BIANCA biophysical model and the FLUKA Monte Carlo transport code, a study was performed to calculate astronaut absorbed doses and equivalent doses following GCR exposure under different shielding conditions. More specifically, the interface with BIANCA allowed us to calculate both the RBE for cell survival, which is related to non-cancer effects, and that for chromosome aberrations, related to the induction of stochastic effects, including cancer. The results were then compared with cancer and non-cancer astronaut dose limits. Concerning the stochastic effects, the equivalent doses calculated by multiplying the absorbed dose by the RBE for chromosome aberrations (“high-dose method”) were similar to those calculated using the Q-values recommended by ICRP. For a 650-day mission at solar minimum (representative of a possible Mars mission scenario), the obtained values are always lower than the career limit recommended by ICRP (1 Sv), but higher than the limit of 600 mSv recently adopted by NASA. The comparison with the JAXA limits is more complex, since they are age and sex dependent. Concerning the deterministic limits, even for a 650-day mission at solar minimum, the values obtained by multiplying the absorbed dose by the RBE for cell survival are largely below the limits established by the various space agencies. Following this work, BIANCA, interfaced with an MC transport code such as FLUKA, can now predict RBE values for cell death and chromosome aberrations following GCR exposure. More generally, both at solar minimum and at solar maximum, shielding of 10 g/cm^2^ Al seems to be a better choice than 20 g/cm^2^ for astronaut protection against GCR.

## 1. Introduction

According to NASA (www.nasa.gov), with the Artemis program, humans will return to the Moon and will establish the first long-term presence on the Earth satellite. Astronauts will be carried from Earth to the lunar orbit (and back) by the Orion spacecraft, which will be launched by “Space Launch System”. The program foresees the construction of an Artemis base camp on the surface, as well as Gateway in the lunar orbit. The latter would be a spaceship with which astronauts will transfer between Orion and the lander, providing a space to live and work and supporting long-term scientific and exploration activities. The subsequent step would consist of sending the first astronauts to Mars. Exploration of the Moon and Mars is intertwined, because the Moon could allow us to test new instruments and equipment that could be used on Mars, including human habitats and life support systems. At the same time, living on Gateway for months could allow researchers to understand how the human body responds in a deep-space environment before committing to the Mars mission, which is expected to last a couple of years.

A human journey to Mars could be characterized by several hazards, which have been classified into groups and are being studied using ground-based analogs and laboratories, as well as the International Space Station (ISS). Radiation is one of the “red risks” reported by a recent NASA study [1] on the highest-priority health problems faced by astronauts due to the well-known health effects, including cancer, cardiovascular disease and cognitive decrements [2]. Other important risks are related to the physiological effects of microgravity [3], as well as the psychosocial effects due to long-term confinement and isolation [4].

When dealing with space radiation effects, it must be taken into account that the space radiation environment is both quantitatively and qualitatively different from that encountered on Earth. In space, astronauts are continuously exposed to galactic cosmic rays (GCRs) consisting of high-energy, charged particles (of which about 87% are protons, 12% are He ions and 1% are heavier ions including iron, which is extremely effective in inducing DNA damage clusters [5]), for which the contribution of high-LET radiation to the total effective dose is substantial. From a quantitative point of view, while the average radiation background on Earth is about 2.4 mSv/year, astronauts on the ISS receive doses in the order of 0.5 mSv/day [6]. In particular, Berger et al. reported a detailed study of long-term dose monitoring onboard Columbus Laboratory of the International Space Station (ISS) performed with different dosimeters within the DOSIS and DOSIS 3D experiments, finding values of up to 0.286 mGy/day with dose equivalent values of 0.647 mSv/day [7]. Higher doses, in the order of 1 mSv/day or more, would characterize a mission to the Moon or Mars, due to the lack of protection against the geomagnetic field. This scenario is further complicated by “Solar Particle Events” (SPEs), which are occasional injections of very high fluxes (up to more than 10^10^ particles·cm^−2^ in few hours) of charged particles coming from the Sun, mainly protons at energies below a few hundred MeV. Such events are very difficult to predict with sufficient notice, although the SPE probability is known to increase with solar activity, which follows an 11-year cycle. However, one should also bear in mind that at solar minimum, when the SPE probability is lower, the GCR flux is higher, due to decreased protection against the Sun’s magnetic field. While the exposure to SPEs is related to early, deterministic effects, continuous exposure to GCRs implies the risk of developing late, stochastic effects, including radiation-induced cancer.

To mitigate this hazard, deep-space vehicles are expected to have significant protective shielding, dosimeters and alerts. Research is also being conducted in the field of medical countermeasures such as radioprotective pharmaceuticals, although none have been approved for astronauts so far [8]. Radioadaptation has also been proposed as a potential way for radiation protection during deep-space travels; a discussion on this topic can be found in [9]. In any case, before sending astronauts on a long-term human mission in deep space, calculations are needed to predict the doses (not only absorbed doses, but also equivalent and effective doses) that the astronauts would receive, and these calculations have to be compared with limits established by the various space agencies.

On this subject, different agencies have different approaches [10]. Concerning the effective-dose limits for stochastic effects, ESA (European Space Agency), RSA (Russian Space Agency) and CSA (Canadian Space Agency) consider a career limit of 1 Sv, independent of astronauts’ age and sex. Recently, NASA changed the approach of adopting age- and sex-dependent limits by establishing a limit of 600 mSv valid for both males and females, independent of age [11,12]. On the contrary, JAXA (Japan Aerospace Exploration Agency) still uses age- and sex-dependent limits. A very detailed discussion can be found in a recent review by Boscolo and Durante [13], and only the main aspects will be summarized herein. Specifically, until 2020, NASA has estimated the dose limits in low Earth orbit (LEO) using the NSCR (NASA Space Cancer Risk) 2012 model [14], also based on the NCRP recommendation of limiting occupational radiation exposure to 3% lifetime excess risk of cancer death [15]. NCRP also outlined that due to the large uncertainties affecting the estimation of heavy-ion health risks, exposure limits for long-term missions in deep space cannot be established until further information is obtained [16]; thus, mission safety can only be predicted within a confidence interval (CI). NASA, therefore, adopted sex- and age-dependent career limits based on 3% REID (risk of exposure-induced death) within a 95% CI. The resulting dose limits for missions of up to 1 year are reported in Table 1 [17]. However, in 2021, National Academy of Sciences (NAS) [18] suggested to use a sex- and age-independent limit obtained by applying the NSCR 2012 model to the most susceptible case (30-year-old female), which corresponds to 0.6 Sv. Although NASA is implementing this new limit, several concerns have been raised by the scientific community, especially because it does not take into account the uncertainties [19].

Analogous to NASA, JAXA is following the NCRP 132 recommendation consisting of basing limits on 3% excess risk of cancer mortality [15]. The calculation of the radiation risk for Japanese astronauts relies on lifetime cancer mortality (LCM), which in turn makes use of cancer mortality values derived from the A-bomb survivor database. The JAXA dose limits are set to 3% LCM [20]; Table 2 reports the effective-dose career limits (in Sv).

Concerning non-cancer effects, all agencies have established limits for skin, eye and blood-forming organs (BFOs). Furthermore, NASA also considers heart and central nervous system (CNS), whereas JAXA includes testes. The limits for missions of different duration are reported in Table 3 (NASA), Table 4 (ESA and RSA) and Table 5 (JAXA). While NASA limits are expressed in Gy-Eq (except for CNS damage by ions with Z > 9, for which the limits are in Gy), the others are in Sv.

Since, in space research, a major uncertainty is related to the RBE of space radiation [21], in this work, we applied the BIANCA biophysical model [22,23] to calculate the RBE of galactic cosmic rays for the induction of both chromosome aberrations, which are indicators of cancer risk [24,25,26,27,28,29], and cell death, which is related to early, deterministic effects. Analogous to previous works where we evaluated the RBE of cancer hadrontherapy beams [30,31,32,33], BIANCA was interfaced with the FLUKA Monte Carlo radiation transport code (www.fluka.org) [34,35,36,37], which is a multi-particle, multi-purpose code applied in a variety of fields, including medical physics, cosmic ray studies, shielding, dosimetry, radiation protection, calorimetry, detector simulation, accelerator-driven systems, etc. In addition to absorbed doses, equivalent doses for a typical Mars mission were calculated in a water phantom exposed to GCRs under different shielding conditions, and the results were discussed with respect to the current astronaut limits.

## 2. Materials and Methods

### 2.1. Simulation of GCR Irradiation Using the FLUKA Transport Code

The physical aspects of irradiation were simulated by means of the FLUKA Monte Carlo transport code. A spherical water phantom with a 15 cm radius was placed into cylindrical aluminum shielding of 38 cm in radius, 180 cm in height and variable thickness. The values considered for the Al thickness were 0.3 and 1 g/cm^2^ (light and nominal spacesuit), 2 and 5 g/cm^2^ (light and nominal spacecraft), 10 and 20 g/cm^2^ (storm shelters to be used in case of intense SPEs). The space between the shielding structure and the water phantom was filled with air.

Concerning galactic cosmic rays, an isotropic spherical source of 200 cm in radius was implemented, and the GCR spectra were based on the model developed by Badhwar and O’Neill [38], which considers elements from Z = 1 to Z = 28. The calculations were performed both at solar minimum (solar modulation parameter ϕ = 465 MV) and at solar maximum (ϕ = 1440 MV). The H, He, C and Fe components of the GCR flux in both cases are shown in Figure 1.

Afterwards, absorbed doses, equivalent doses and the corresponding relative biological effectiveness (RBE) values were calculated. While the equivalent doses to different tissues/organs were calculated using the Q-values reported in ICRP publication 60 [39], the RBE calculation was based on the BIANCA biophysical code, as described in the next section. Briefly, two radiobiological databases generated with BIANCA were considered: the first one describing human skin fibroblast (HSF) cell death, the second one describing the induction of dicentric chromosomes in blood lymphocytes. Both databases consist of linear and quadratic coefficients allowing dose–response to be predicted as a function of the ion type (1 ≤ Z ≤ 26) and LET. While cell death is mainly related to the induction of early, deterministic damage, lymphocyte aberrations are regarded as representative of late stochastic damage, typically cancer. As a consequence, the cell survival RBE was used to calculate the equivalent dose (in Gy-Eq) for deterministic effects (not applicable to cataract induction; ICRP Publication 58 [40]), whereas the RBE for lymphocyte dicentrics was used to calculate the equivalent and the effective doses (in Sv) for stochastic effects.

In this work, we considered 650 days in free space to represent a “short-stay” Mars mission, which would consist of 620 days in free space and 30 days on Mars’ surface [41]. Due to the short time spent on the planet surface, the difference between the Mars radiation environment and free space was neglected.

### 2.2. RBE Calculation by Means of the BIANCA Biophysical Model

BIANCA is a biophysical model, implemented as a Monte Carlo code, that simulates the induction of cell death and chromosome aberrations following cell irradiation with photons and with different monochromatic ion beams, that is, with different ion types and different energy values. The assumptions and parameters of the model, as well as the main simulation steps to obtain (simulated) dose–response curves for chromosome aberrations or cell death, have been discussed in detail in several publications [22,23]. Herein, we will just mention that BIANCA relies on the idea that ionizing radiation induces, in the cell nucleus, a certain yield of DNA “critical lesions”, which interrupt the continuity of the chromatin fiber producing independent chromatin fragments. These fragments either remain un-rejoined, or undergo distance-dependent incorrect rejoining (i.e., rejoining with the “wrong” partner), giving rise to different chromosome aberration types [42]. Finally, some aberration types (dicentric chromosomes, rings and deletions) are assumed to lead to (clonogenic) cell death.

With the goal of predicting cell survival curves for different cell types as a function of radiation type and energy, as a first step, a radiobiological database describing the survival of V79 cells (chosen as a reference) as a function of dose, ion type and energy has been produced [43]; afterwards, an algorithm has been developed to predict survival curves for other cell types [44]. Concerning chromosome aberrations, BIANCA has been applied to obtain a database consisting of linear and quadratic coefficients describing dose–response curves for dicentric chromosomes in circulating lymphocytes. First, such work has been performed for ions with Z between 1 (protons) and 8 (oxygen), thus allowing for applications in the field of cancer hadrontherapy [43]. More recently, heavier ions up to Fe have been implemented, thus making it possible to perform calculations for space radiation [44,45].

The construction of the database describing HSF cell survival has been described in detail in [44,45]. Herein, it is sufficient to mention that such database consists of a table that, for each ion type and LET, reports the linear and quadratic coefficients (*α* and *β*, respectively) characterizing the well-known equation that is usually adopted to describe cell survival dose–response, i.e.,
(1)$$S\left(D\right)=e^{-\alpha D-\beta D^{2}}$$
where S(D) is the fraction of surviving cells after receiving an absorbed dose *D*.

The BIANCA database for lymphocyte dicentrics, described in detail in previous works [43,44,45], contains, again, for each ion type and LET value, the linear and quadratic coefficients (*a* and *b*, respectively) characterizing a typical dicentric dose–response curve at low and intermediate doses, which can be described as follows:(2)$$Y\left(D\right)=aD+bD^{2}$$
where Y(D) is the mean number of dicentrics per cell after an absorbed dose *D*.

In principle, these tables can be read by any radiation transport code; in this work, they were read by FLUKA, exploiting a pre-existing interface between the two codes and taking into account that space radiation is a mixed field involving different particles and different energy values. Several ways exist to deal with the effects of mixed fields [46]; in this work, whenever according to FLUKA a certain amount of energy (and thus a certain dose, *D_i_*) was deposited in a target voxel by a given radiation type *i* (where “radiation type” means a given particle of given energy, and thus given LET), FLUKA read from the tables the corresponding coefficients (*α_i_* and *β_i_* for cell survival; *a_i_* and *b_i_* for lymphocyte dicentrics) and used them to calculate the average coefficients describing the fraction of surviving cells (called *α* and *β*) or those describing the yield of dicentrics (called *a* and *b*), due to the mixed field in that voxel.
(3)$$\alpha =\frac{\sum_{i=1}^{n}\alpha_{i}D_{i}}{\sum_{i=1}^{n}D_{i}}$$
(4)$$\sqrt{\beta }=\frac{\sum_{i=1}^{n}\sqrt{\beta_{i}}D_{i}}{\sum_{i=1}^{n}D_{i}}$$
(5)$$a=\frac{\sum_{i=1}^{n}a_{i}D_{i}}{\sum_{i=1}^{n}D_{i}}$$
(6)$$\sqrt{b}=\frac{\sum_{i=1}^{n}\sqrt{b_{i}}D_{i}}{\sum_{i=1}^{n}D_{i}}$$

Finally, the RBE in each voxel was calculated as $\frac{D_{x}}{D}$, where $D_{}$ is the total absorbed dose in the voxel, whereas $D_{x}$ is the photon dose needed to obtain the same effect. The value of $D_{x}$ for cell survival was calculated according to Equation (7), whereas that for dicentrics was based on Equation (8).
(7)$$D_{X}=\frac{\left[-\alpha_{X}+\sqrt{\alpha_{X}^{2}+4\beta_{X}lnS}\right]}{2\beta_{X}}$$
(8)$$D_{X}=\frac{\left[-a_{X}+\sqrt{a_{X}^{2}+4b_{X}Y}\right]}{2b_{X}}$$

To take into account the effects of low doses and low dose rates characterizing the exposure to cosmic rays, a single and maximum value of the RBE is required, defined as the RBE at minimal doses ($RBE_{M}$) and calculated as the ratio between the initial slopes of the dose–effect curves for the radiation under study and the reference radiation [47].

For stochastic effects, according to ICRP Publication No. 92 [47], two methods can be applied to determine *RBE_M_*: the “low-dose” method and the “high-dose” method. According to the low-dose method, *RBE_M_* is calculated as follows:(9)$$RBE_{M}=\frac{\alpha }{\alpha_{\gamma }}$$
where α and $\alpha_{\gamma }$ are the initial slopes of the dose–effect curves for GCRs and gamma rays, respectively. The $RBE_{M}$ values calculated using this method are highly dependent on the considered reference radiation, as well as their responses at low doses and low dose rates. The low effectiveness of low doses or low dose rates of *γ*-rays, as well as the uncertainties due to differences in the linear slopes derived from acute irradiations compared with low-dose-rate experiments, can lead to large $RBE_{M}$ values.

To remove these uncertainties, Committee on Interagency Radiation Research and Policy Coordination (CIRRPC) suggested to adopt the high-dose method, which uses the observed high-dose RBE, called $RBE_{H}$, and extrapolates it to low doses by means of the “dose and dose-rate reduction effectiveness factor” (DDREF) [39], which is inferred from the entirety of the data that are relevant to late effects in humans. According to the high-dose method, the following equation holds:(10)$$RBE_{A}=DDREF\cdot RBE_{H}$$
where $RBE_{A}$ is the $RBE_{M}$ calculated with the high-dose method, DDREF is the dose and dose-rate reduction effectiveness factor, and $RBE_{H}$ is the $RBE_{}$ value at high doses. In line with the choice reported in ICRP Publication No. 60 [39] and with the subsequent recommendations published by NRPB [48], in the present work, we selected a DDREF value of 2.

Concerning the calculation of $RBE_{H}$, National Radiobiological Protection Board (NRPB) has defined it as the ratio of the initial slope of the dose–response curve for cancer induction by high-LET radiation to the slope of the linear fit to intermediate- and high-dose data for cancer induction by low-LET radiation. Since space radiation is a complicated mixture of high- and low-LET radiation, in this work, we re-defined $RBE_{H}$ as the ratio of the slope of the linear fit to intermediate- and high-dose data for dicentric induction by GCRs to the slope of the linear fit to intermediate- and high-dose data for dicentric induction by low-LET radiation.
(11)$$RBE_{H}=\frac{\frac{dY\left(D\right)}{dD}}{\frac{dY_{\gamma }\left(D_{\gamma }\right)}{dD_{\gamma }}}=\frac{a+2bD}{a_{\gamma }+2b_{\gamma }D_{\gamma }}$$
where Y(D) is the dicentric yield for GCRs, whereas $Y_{\gamma }\left(D_{\gamma }\right)$ is the dicentric yield for gamma rays. For these calculations, a value of *D_γ_* = 1 Gy was considered, according to [49].

Concerning deterministic effects, the RBE values can be obtained at doses corresponding to the threshold level for individual deterministic effects. This task is complex because there are different RBE values in different tissues for different endpoints; moreover, the threshold doses vary among individuals and are not always easily determined. ICRP Publication No. 58 [40] recommends referencing the low-dose limit of the RBE for deterministic effects, although this entails the extrapolation to doses at which the responses to both the considered radiation and the reference radiation are below the threshold. In this work, we calculated the $RBE_{M}$ value for deterministic effects such as in the case of the low-dose method for stochastic effects, that is, by considering the ratio of the initial slope of the dose–effect curve for GCRs to that for gamma rays, respectively.

## 3. Results and Discussion

### 3.1. Absorbed Dose, Equivalent Dose and RBE

Figure 2 shows the average absorbed doses in the phantom (in mGy/day) and the corresponding equivalent doses (in mSv/day; calculated using the Q-values reported in ICRP publication No. 60 [39]) obtained by FLUKA as a function of the Al shielding thickness in the case of GCR exposure at solar minimum (ϕ = 465 MV). According to these results, the absorbed dose per day remained approximately constant (0.4 mGy/day) with the increase in shielding in the range of 0–10 g/cm^2^, whereas a slight increase was observed at 20 g/cm^2^; this may have been due to an increased role of secondary particles produced in the shielding itself, as found in a previous work [50]. On the contrary, the equivalent dose showed a decrease from 1.5 mSv/day to 1.05 mSv/day in the range of 0–10 g/cm^2^, followed by a slight increase at 20 g/cm^2^ due to the increase in the absorbed dose. These numbers are consistent with the results obtained by several other authors, which of course depends on the adopted methods, shielding conditions, etc. In summarizing these results for a Mars mission at solar minimum, Simonsen et al. [41] report a range of 300–450 mGy for the absorbed dose and 870–1200 mSv for the equivalent dose.

The dependence of the (total) absorbed and equivalent doses on the shielding thickness can be explained by taking into account the contributions of the different particles, both primary and secondary ones. To clarify this issue, Figure 3 shows such contributions for two sample shielding cases, that is, no shielding (panel a) and 5 g/cm^2^ shielding (panel b). For instance, as reported in Figure 3a, without shielding, ~42% of the total equivalent dose was due to particles with Z between 1 and 8 and 54% to particles with Z between 9 and 28. On the contrary, Figure 3b shows that, with 5 g/cm^2^ Al shielding, these numbers were 57% and 40%, respectively. This reflects the fact that an increase in shielding implied an increase in projectile fragmentation and thus an increased role of (secondary) ions that have lower Z but approximately the same velocity as the primary ion, thus lower LET and biological effectiveness. In turn, this implied a decrease in the equivalent dose, whereas the absorbed dose remained roughly constant. However, as mentioned above, with very large shielding, such as 20 g/cm^2^ or more, the doses tended to increase again.

Although the figures reported in this paper refer to the solar minimum, which is the most critical condition with respect to the risks related to GCR exposure, analogous calculations were also performed at solar maximum, for which the values of absorbed and equivalent doses were found to be approximately one third of the values at solar minimum.

To help to interpret the results that are reported in Section 3.2 and Section 3.3, Figure 4 shows the dependence of the RBE (both for lymphocyte dicentrics and for cell survival) on the absorbed dose, obtained by interfacing FLUKA with BIANCA. The results are shown for the sample case of 5 g/cm^2^ Al shielding at solar minimum. As expected, both for cell survival and for lymphocyte dicentrics, the RBE increased with the decrease in the dose, and a maximum value (RBE_M_) was reached at minimal doses. However, such RBE variation was much more pronounced for lymphocyte dicentrics; while, for cell survival, the difference between RBE_M_ and the RBE at 0.5 Gy was 10%, for lymphocyte dicentrics, RBE_M_ was approximately three times the RBE at 0.5 Gy.

This can be explained by taking into account that when compared with photon cell survival curves, a typical photon dose–response for lymphocyte dicentrics is characterized by a much more pronounced curvature and thus a lower alpha/beta ratio. In turn, here, this implied a larger difference between the dicentric RBE at low doses, which was much higher than the RBE for cell survival, and the dicentric RBE at higher doses, which was similar to the cell survival RBE.

### 3.2. Stochastic Effects (Cancer)

In view of a comparison with the astronaut limits for stochastic effects (cancer), Figure 5 shows the calculated values of the following quantities: Q (obtained as the ratio between the equivalent and absorbed doses reported in Figure 2); RBE_M_ for lymphocyte dicentrics (obtained by applying Equation (9)); RBE_A_ for lymphocyte dicentrics (obtained by applying Equations (10) and (11)). The obtained values are shown as a function of the Al shielding thickness at solar minimum. Although Q is systematically higher than RBE_A_ (especially with small shielding, whereas the difference tends to become negligible with larger shielding), the RBE_A_ values are rather close to the Q-values. On the contrary, the RBE_M_ values are much higher (by approximately a factor 2) than both RBE_A_ and Q. This may be explained by taking into account that at very low doses, the linear component of the dicentric dose–response was extremely small; however, considering that this value is affected by large uncertainties, the use of RBE_A_ seems preferable to RBE_M_, as also suggested by other authors [51].

Table 6 shows the equivalent doses obtained by multiplying the absorbed doses (in mGy) either by Q, by the dicentric RBE_A_ or by the dicentric RBE_M_ in the case of 650 days in free space at solar minimum. As a consequence of the scenario shown in Figure 5, the values obtained using the RBE_A_ approach are rather close to those obtained using Q, although the latter are systematically higher, especially with small shielding. On the contrary, the values obtained using the RBE_M_ approach are much higher than those obtained using the other two approaches. In particular, if one adopted the RBE_M_ approach, even in the “best” case (10 g/cm^2^ Al shielding), the dose would be 1078 mGy·RBE, which is higher than the career limits established by all space agencies.

Concerning Q and RBE_A_, both approaches provided values that are consistent with the ranges reported by Simonsen et al. [41], which are 870–1200 mSv and 550–800 mGy-Eq. Concerning the comparison with the (stochastic) limits, both approaches would make it possible to respect the 1 Sv career limit adopted by ESA and RSA. With respect to the age- and sex-dependent limits adopted by NASA until 2020 (see Table 1), our calculations show that with 10 g/cm^2^ Al shielding, both the RBE_A_ approach and the Q approach implied that the limit for 30-year-old females (600 mSv) would not be respected. Interestingly, with 5 g/cm^2^ Al shielding, the RBE_A_ approach (which resulted in 659 mGyEq) implied that only the limit for 30-year-old females would not respected, whereas with the Q approach (which resulted in 729 mSv) also the limit for 40-year-old females would not be respected. This example shows that even the choice of using Q-values or RBE_A_ values influences the calculation outcomes and, possibly, the crew member selection. However, by applying the current NASA limit of 600 mSv, according to the present calculations, no NASA astronaut could participate in a 650-day mission at solar minimum without exceeding the limit.

Table 7 reports the same kind of information in the case of solar maximum. Again, the values obtained using the RBE_A_ approach are rather close to those obtained using Q, although the latter are systematically higher. Moreover, the values obtained using RBE_M_ are much higher than those obtained using the two other approaches. In any case, even with the RBE_M_ approach, all values are always below the limits established by the various space agencies, since at solar maximum, the GCR absorbed doses are much lower than those absorbed at solar minimum.

### 3.3. Non-Cancer Effects

In view of the comparison with the limits for deterministic effects, Figure 6 reports the values of RBE_M_ and RBE at 0.5 Gy (RBE_0.5_) for cell survival, for different shielding thicknesses at solar minimum. As expected, RBE_M_ is higher than RBE_0.5_ for each considered thickness; however, these differences do not exceed 12%. Following the ICRP recommendations reported in Publication No. 58 [40], i.e., considering the low-dose limit of the RBE for deterministic effects, although considering doses below the threshold to be somewhat artificial, using RBE_M_ is a useful practical approximation, also considering that the overestimation does not exceed 12%.

Table 8 shows the values obtained by multiplying the absorbed dose (in mGy) by the RBE_M_ value for cell survival shown in Figure 6. The results are shown for a period of 7, 30, 365 or 650 days, where the latter corresponds to a short-stay Mars mission. As expected, due to the low dose rate of cosmic rays, even for a Mars mission, almost all calculated values are largely below the limits for deterministic effects established by the various space agencies, as reported in Table 3, Table 4 and Table 5. Also for deterministic effects, an Al shielding of 10 g/cm^2^ resulted to be the most protective one among the different considered values. In fact, the doses found for 20 g/cm^2^ are higher than those found for 10 g/cm^2^ due to the increase in the absorbed dose already discussed above.

## 4. Conclusions

This work shows that the BIANCA biophysical model, when interfaced with a radiation transport code, can calculate the RBE of galactic cosmic rays both for cell survival and for chromosome aberrations in blood lymphocytes. While the RBE for cell survival was applied to calculate equivalent doses with respect to non-cancer effects, the lymphocyte dicentric RBE was applied for stochastic effects (typically, cancer), obtaining values that are in line with those obtained using the radiation quality factors reported in ICRP 60 [30]. Since the latter are practical quantities for radiation protection purposes, this work suggests that the current estimates based on quality factors may be integrated with analogous calculations based on RBE values for the induction of some effects related to cancer induction, such as lymphocyte aberrations, which may be more realistic, since they are more closely related to the underlying radiobiological mechanisms. Furthermore, this work confirms that the use of RBE_A_ should be preferred to that of RBE_M_, which is affected by large uncertainties. On the contrary, this work confirms that for deterministic effects, RBE_M_ seems more suitable than RBE_0.5_.

Concerning the astronaut limits, our results confirm that stochastic effects, in particular cancer, represent the main concern for GCR exposure, since the limits for non-cancer effects would be respected even in a 650-day Mars mission at solar minimum. According to these calculations, in this scenario, 10 g/cm^2^ Al shielding would allow the 1 Sv career limit adopted by ESA and RSA, as well as most of the age- and sex-dependent limits established by JAXA, to be respected; possible exceptions would be the JAXA limits for males younger than 30 years old and females younger than 40. However, by applying the NASA limit of 600 mSv, according to the present calculations, no NASA astronaut could participate in a 650-day mission at solar minimum without exceeding the limit. More generally, both at solar minimum and at solar maximum, shielding of 10 g/cm^2^ Al seems to be a better choice than that of 20 g/cm^2^ for astronaut protection against GCRs.

## Figures and Tables

**Figure 1 ijms-24-02328-f001:**
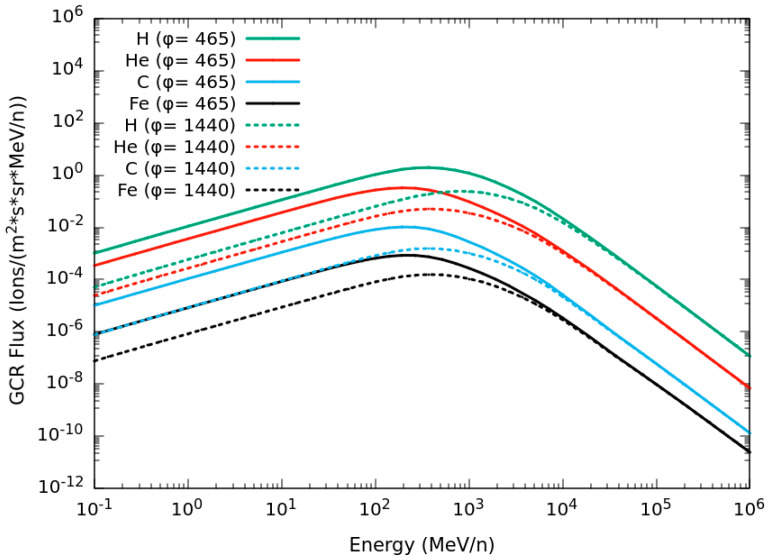
H, He, C and Fe components of the GCR flux at ϕ = 465 (solid lines; solar minimum) and ϕ = 1440 (dashed lines; solar maximum).

**Figure 2 ijms-24-02328-f002:**
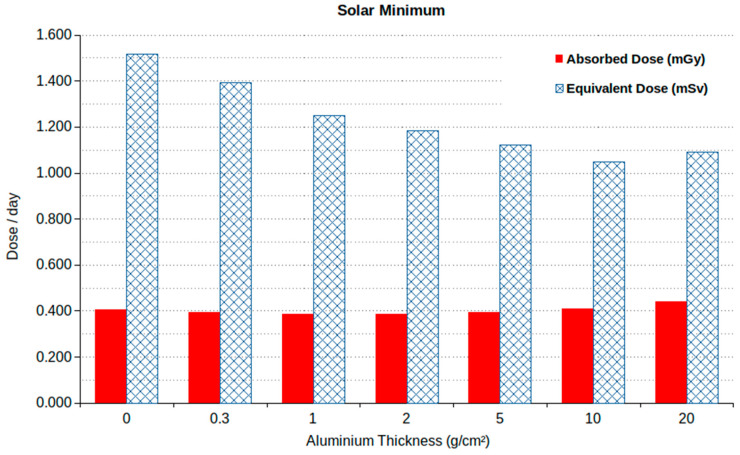
Absorbed and equivalent doses (in mGy/day and mSv/day, respectively) at solar minimum as a function of Al shielding thickness. (The error bars are not visible because the relative error is below 1%.)

**Figure 3 ijms-24-02328-f003:**
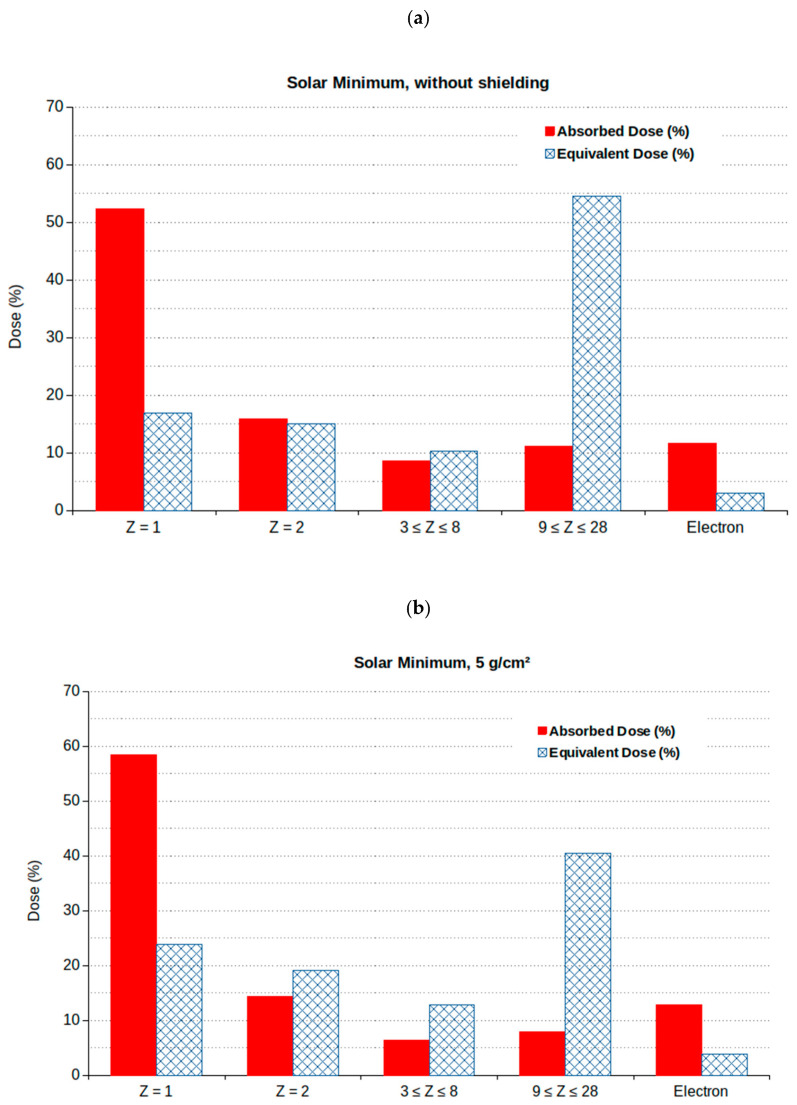
Percentage contribution to the total dose of the different GCR components at solar minimum in case of no shielding (panel **a**) or 5 g/cm^2^ shielding (Al shielding thickness) (panel **b**).

**Figure 4 ijms-24-02328-f004:**
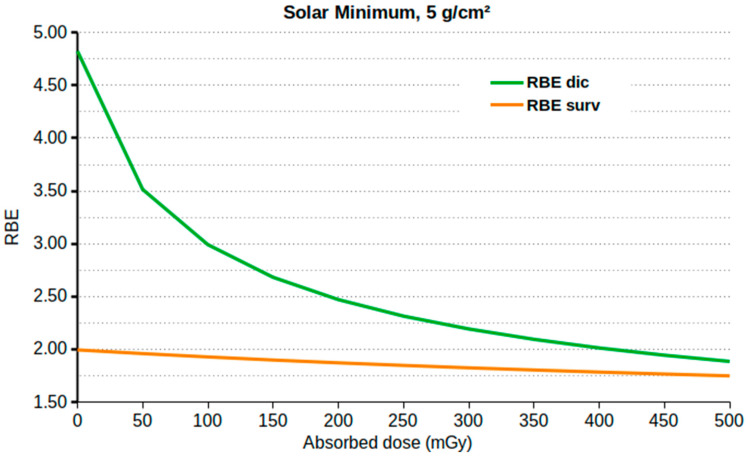
RBE for lymphocyte dicentrics (**upper** line) and cell survival (**lower** line) as a function of the absorbed dose for the sample case of 5 g/cm^2^ Al shielding at solar minimum.

**Figure 5 ijms-24-02328-f005:**
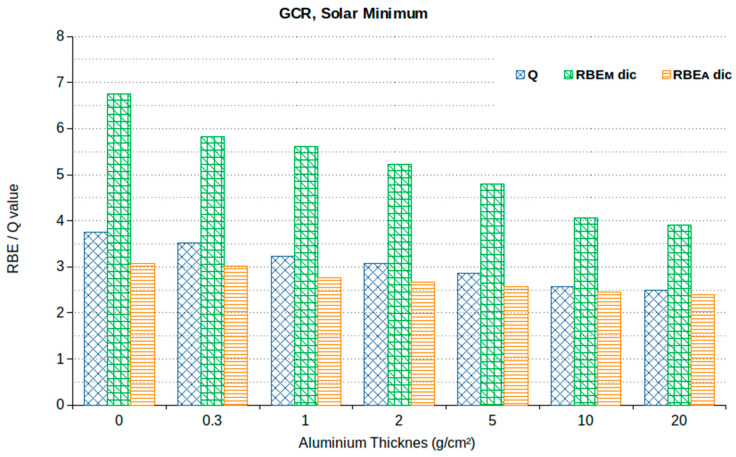
Calculated values of Q, dicentric *RBE_M_* and dicentric *RBE_A_* with different Al shielding thicknesses at solar minimum.

**Figure 6 ijms-24-02328-f006:**
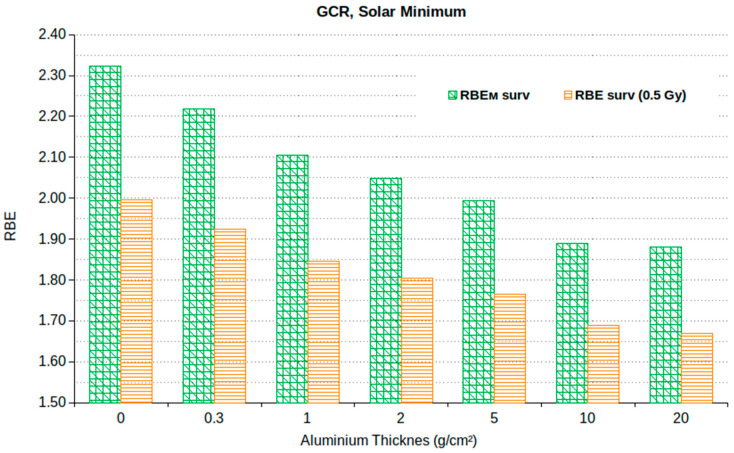
RBE_M_ and RBE at 0.5 Gy for cell survival at solar minimum.

**Table 1 ijms-24-02328-t001:** Effective-dose career limits based on the NASA Space Cancer Risk 2012 model for 1-year missions [17].

Age	Male (Sv)	Female (Sv)
30	0.78	0.60
40	0.88	0.70
50	1.00	0.82
60	1.17	0.98

**Table 2 ijms-24-02328-t002:** Effective-dose career limits established by JAXA [20].

Age at First Space Flight	Males (Sv)	Females (Sv)
27–30	0.60	0.50
31–35	0.70	0.60
36–40	0.80	0.65
41–45	0.95	0.75
>45	1.00	0.80

**Table 3 ijms-24-02328-t003:** NASA limits for non-cancer effects [12].

	30-Day Limit (Gy-Eq)	1-Year Limit (Gy-Eq)	Career Limit (Gy-Eq)
Skin	1.5	3.0	6.0
Eye	1.0	2.0	4.0
BFO	0.25	0.5	N.A.
Heart	0.25	0.5	1.0
CNS	0.5	1.0	1.5
CNS (Z > 9)	N.A.	0.10 Gy	0.25 Gy

N.A., not applicable.

**Table 4 ijms-24-02328-t004:** ESA and RSA * limits for non-cancer effects [10].

	30-Day Limit (Sv)	1-Year Limit (Sv)
Skin	1.5	3.0
Eye	0.5	1.0
BFO	0.25	0.5

* In addition to the limits reported in this table, RSA also established career limits for eye and skin (2.0 and 6.0 Sv, respectively), as well as an acute (one-time) limit of 0.15 Sv for BFOs.

**Table 5 ijms-24-02328-t005:** JAXA limits for non-cancer effects [10].

	1-Week Limit (Sv)	1-Year Limit (Sv)	Career Limit (Sv)
Skin	2.0	7.0	20.0
Eye	0.5	2.0	5.0
BFOs	N.A.	0.5	N.A.
Testes	N.A.	1.0	N.A.

N.A., not applicable.

**Table 6 ijms-24-02328-t006:** Equivalent doses for 650 days in free space at solar minimum, calculated by multiplying the absorbed doses (in mGy) either by Q (column 2), by the dicentric RBE_A_, (column 3) or by the dicentric RBE_M_ (column 4).

Al Thickness (g/cm^2^)	Equivalent Dose (mSv)	Equivalent Dose (mGy·RBE_A_)	Equivalent Dose (mGy·RBE_M_)
0	986.7	809.8	1777.4
0.3	904.5	774.3	1496.0
1	812.1	693.5	1413.3
2	770.4	669.3	1312.7
5	729.0	658.9	1228.2
10	681.6	652.2	1077.7
20	708.5	680.9	1106.4

**Table 7 ijms-24-02328-t007:** Equivalent doses for 650 days in free space at solar maximum, calculated by multiplying the absorbed doses (in mGy) either by Q (column 2), by the dicentric RBE_A_, (column 3) or by the dicentric RBE_M_ (column 4).

Al Thickness (g/cm^2^)	Equivalent Dose (mSv)	Equivalent Dose RBE_A_ Dic (mGy·RBE_A_)	Equivalent Dose RBE_M_ Dic (mGy·RBE_M_)
0	240.9	211.5	426.2
0.3	249.2	217.3	442.8
1	279.5	238.3	515.8
2	319.6	253.1	579.4
5	254.1	222.2	432.6
10	227.6	222.3	358.1
20	266.4	255.4	389.9

**Table 8 ijms-24-02328-t008:** Equivalent doses (mGy·RBE_M_) at solar minimum calculated by multiplying the absorbed doses by the RBE_M_ value for cell survival.

Al Thickness (g/cm^2^)	7 Days	30 Days	365 Days	650 Days
0	6.58	28.19	342.94	610.71
0.3	6.16	26.30	320.00	569.87
1	5.67	24.44	297.36	529.55
2	5.53	23.74	288.84	514.38
5	5.46	23.52	286.17	509.62
10	5.39	23.11	281.23	500.81
20	5.74	24.62	299.50	533.36

## Data Availability

The data were obtained by the BIANCA and FLUKA simulation codes.

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
