# Peer review of "A Mission to Mars: Prediction of GCR Doses and Comparison with Astronaut Dose Limits"

_ijms, 2023, doi:10.3390/ijms24032328_

Round 1
Reviewer 1 Report
Presence of radiation is universal; be it on earth or extraterrestrials locations including moon and Mars. Artemis program - exploration of mission to Mars: where the astronauts planned to work and supporting long-term scientific and exploration activities, including human habitats. Excitingly there to astronauts exposed continuous Galactic Cosmic Rays (GCR); the environment is rich is of high-energy charged particles (about 87% protons, 12% He ions 57 and 1% heavier ions in fluence). All the high-LET radiation and its contribution to the total effective dose are substantial; un-doughtly it could enhance the risk for both stochastic as well as tissue reactions. Nevertheless, it is highly challenging to predict the human response to those mixed filed high flux radiation field. Thus an exciting filed of research..
In view of those encounters, the investigators adopted a progarmme the (BIANCA biophysical model and FLUKA Monte-Carlo transport code) to predict the dose, and RBE in terms of radiation specific response end point (Dicentric chromosomes and cell survival Clonogenic assay) following GCR exposure under different shielding conditions. It was suggested that both at solar minimum and at solar maximum, a shielding of 10 g/cm2 aluminum thickness would be a better choice than 20 g/cm2 for astronauts’ protection from those health effects (including cardiovascular system related) against GCR.
Author Response
We thank reviewer 1 for his/her comments, which do not imply any change in the manuscript
Reviewer 2 Report
This is an interesting article testing the biophysical model BIANCA in RBE calculations for Galactic Cosmic Rays. It is sound and straightforward work showing, that indeed BIANCA provides estimates in line with those reported by ICRP.
Authors start with a comprehensive introduction providing background to space weather and radiation protection regulations. And here are following discrepancies:
1. Introduction misses important references, for example:
- line #51 - references are missing for other than cardiovascular effects.
- line #52
Please add those above mentioned and other missing references.
2. Line #59 - Authors provide estimations of ISS radiation dose. There are physical measurements (e.g. DOSIS experiments) that report slightly higher doses. Please provide information on physical measurements of radiation dose onboard of ISS.
3. Authors, starting in the introduction, and throughout the entire text, seem to mix NASA regulations for radiation protection. For example they provide contradictory information on NASA regulations in lines #80-83 and 95-97. Furthermore, Authors provide outdated information on NASA regulations (line #359). Current NASA limit is 600 mSv (3% REID for a 35 y.o. female) which is applied to both: females and males, so it is age and sex independent. Also Line #354 and #357 provides reference to old NASA limit
Please look into and quote following references:
Space Radiation and Astronaut Health: Managing and Communicating Cancer Risks |The National Academies Press
NASA regulations: 2022-01-05-NASA-STD-3001-Vol1-Rev-B-Final-Draft-Signature-010522.pdf
4. Table 1 – reference is missing – please update reference.
5. Tables 3, 4 and 5 have no references. Please add references.
6. Table in line #117 has no number, no description, and no reference. Please add Number, description, and reference.
7. Authors use symbols, that are not explained, for example "Ï•". Please make sure that all acronyms and symbols are explained.
Reviewer 3 Report
The presented manuscript is well written and describes important topic. I have no strong comments or remarks. I generally recommend it to be published, but three items seem to be discussed in deeper way:
1) dose limits and their dependence on different factors: in this context Authors can comment / discuss the problem of individual radioresistance and radioadaptation of astronauts. There is some number of scientific papers about that topic;
2) mixed radiation field: also in this context one can find many research articles about dose assessment in mixed radiation field, with different types of radiation, with different LET;
3) FLUKA and BIANCA codes are generally known worldwide - however, few additional sentences with more information / details about both codes are more than welcome.
